# Outcomes of a QST Protocol in Healthy Subjects and Chronic Pain Patients: A Controlled Clinical Trial

**DOI:** 10.3390/biomedicines11041023

**Published:** 2023-03-27

**Authors:** Patrícia Dias, Isaura Tavares, Sara Fonseca, Daniel Humberto Pozza

**Affiliations:** 1Department of Biomedicine, Faculty of Medicine, University of Porto, 4200-319 Porto, Portugal; 2Institute for Research and Innovation in Health and IBMC, University of Porto, 4200-135 Porto, Portugal; 3Anesthesiology Department, São João University Hospital Centre, 4200-319 Porto, Portugal; 4Department of Histology, Faculty of Nutrition and Food Sciences, University of Porto, 4150-177 Porto, Portugal

**Keywords:** chronic pain, quantitative sensory testing, osteoarthritis, preoperative pain, heat pain threshold, analgesia, opioid misuse, quality of life

## Abstract

Chronic pain is an important cause of disability with a high burden to society. Quantitative sensory testing (QST) is a noninvasive multimodal method used to discriminate the function of nerve fibers. The aim of this study is to propose a new, reproducible, and less time-consuming thermal QST protocol to help characterize and monitor pain. Additionally, this study also compared QST outcomes between healthy and chronic pain subjects. Forty healthy young/adult medical students and fifty adult/elderly chronic pain patients were evaluated in individual sessions including pain history, followed by QST assessments divided into three proposed tests: pain threshold, suprathreshold, and tonic pain. In the chronic pain group, a significantly higher pain threshold (hypoesthesia) and a higher pain sensibility (hyperalgesia) were demonstrated at threshold temperature when compared to healthy participants. The sensitivity to the suprathreshold and tonic stimulus did not prove to be significantly different between both groups. The main results demonstrated that the heat threshold QST tests can be helpful in evaluating hypoesthesia and that the sensitivity threshold temperature test can demonstrate hyperalgesia in individuals with chronic pain. In conclusion, this study demonstrates the importance of using tools such as QST as a complement to detect changes in several pain dimensions.

## 1. Introduction

Pain is the main symptom leading patients to seek medical care, and chronic pain is one of the main sources of “years of lived with disability”, representing an enormous economic burden to health care systems [1,2,3,4]. More than 30% of the world’s population suffers from chronic pain [2,5,6,7,8]. Chronic pain lacks the protective function of acute pain and may be caused by several conditions, such as long-term inflammatory events, traumatic and metabolic injuries of the nervous system, and diseases or their treatment, such as cancer and chemotherapy [1,2,4]. Among the most prevalent types of chronic pain, osteoarthritis (OA) is one of the most debilitating diseases and the main cause of poor quality of life in the elderly [9]. The mechanisms underlying the persistence of pain in OA are complex, with peripheral and central sensitization mechanisms. The former has been proposed as a target of new pharmacological therapies (e.g., TNF-α antibodies, such as adalimumab and infliximab) yet with inconsistent results, and treatment of OA pain remains a clinical challenge [4,9,10,11]. Frequently patients with OA are referred for surgery to replace the damaged joint.

Quantitative sensory testing (QST) is a simple, quick, and noninvasive psychophysical test that assesses the functions of the somatosensory system, targeting small sensory fibers [12,13,14]. QST can assess a variety of sensory modalities and stimuli, such as thermal, pressure, light touch, vibration, or chemical stimulus, and may be applied to different tissues [12,15]. The feasibility of QST protocols remains a challenge since QST requires the active collaboration of the subject/patient and an attentive trained operator [16]. QST has been proven to be sensitive and useful for studying a wide variety of diseases, such as Parkinson’s disease, major depressive disorder and hereditary amyloidosis [15,17,18,19], and chronic pain, including diagnosis of small-fiber neuropathy [20,21,22,23] and genetic loci involved in pain sensitivity [24]. The thermal modality is considered the best QST pain assessment because it is associated with small-fiber neuropathies, with 93% of positive predictive value [25,26], even for those that would go unnoticed on routine neurological examination [27].

Preoperative pain sensitivity testing may identify patients who will require individualization of pain management strategy [12,28,29]. QST is a valuable tool for stratifying risk patients according to their sensory phenotype and underlying pain mechanism to provide targeted neuroprotection in vulnerable subpopulations, as well as a tailored analgesic treatment plan [1,25,30,31]. The severity of acute postoperative pain is correlated with the potential development of persistent chronic pain, which can also be predicted using QST [32,33]. This is of paramount importance as, despite the higher incidence, chronic postoperative pain is underrecognized and undertreated [34,35,36]. Furthermore, the prediction of the outcome of relevant interventions aims to reduce acute postoperative pain and possibly reduce the occurrence and/or severity of chronic postoperative pain [37,38].

Nowadays, there are many QST protocols in the literature, some of them with good predictive results [13,39,40,41,42,43,44,45], and others that failed to prove that QST can predict chronic pain, particularly in some modalities or subgroups [46,47,48,49,50,51]. Consequently, there are gaps in the current knowledge that need to be approached with a viable QST protocol for clinical context. Furthermore, most QST studies use lengthy protocols and assess pain sensitivity in groups of patients who suffer from multiple pathologies and are focused on comparing preoperative and postoperative pain sensitivity [16,52,53], lacking studies comparing healthy subjects and patients with chronic pain. The objective of this controlled clinical study is to propose a new, reproducible, and less time-consuming thermal QST protocol to help characterize and monitor pain. Additionally, this study also compared QST outcomes between healthy and chronic pain subjects suffering from OA.

## 2. Materials and Methods

This longitudinal, prospective controlled clinical trial was approved by the Ethics and Research Committee of the São João Hospital (Numbers 293-13 for the chronic pain patients and 374-19 for the healthy volunteers). The study lasted between January 2014 and January 2015 for chronic pain patients and between November 2019 and April 2021 for healthy volunteers.

All sessions were held in an air-conditioned room with a constant temperature of 23 °C and no noise to avoid distractions. Volunteers were asked if they felt comfortable and the tests were carried out by the same investigator. Prior to the testing, participants answered validated questionnaires based on Brief Pain Inventory (BPI) [54,55] for previous pain history, and additional questions intended to briefly evaluate the current psychological status of the volunteer (Appendix A).

### 2.1. Selection of Healthy Volunteers

#### 2.1.1. Inclusion Criteria

Inclusion criteria included a consecutive sample of healthy young medical students free from chronic pain of any type, the ability to understand the purpose and instructions of the study, and the age between 18 and 25 years.

#### 2.1.2. Exclusion Criteria

Exclusion criteria included reported pain, female volunteers during the menstrual cycle, any skin disease or allergy, medication use (except for oral contraceptives), opioid use in the last month, inability or unwillingness to give written informed consent, psychiatric disease, use of analgesics, alcohol, or drug abuse [56,57,58].

### 2.2. Selection of Patients with Chronic Pain

#### 2.2.1. Inclusion Criteria

Inclusion criteria included a consecutive sample of chronic pain patients, followed in the Orthopedics Department of São João Hospital, the ability to understand the purpose and instructions of the study, and the age between 40 and 85 years. All patients presented chronic pain in the knee or hip due to osteoarthritis and were approached in the preoperative period for knee or hip replacement.

#### 2.2.2. Exclusion Criteria

Exclusion criteria included allergy to analgesics, ASA status higher than III, history of peptic ulcer disease or gastrointestinal bleeding, use of strong opioids in the last month, inability or unwillingness to give written informed consent, those unable to communicate, psychiatric disease, pre-eclampsia or eclampsia, significant pulmonary disease, intraoperative complications, modified surgical procedure, or deviations from the standardized anesthetic regimen, alcohol, or drug abuse [56,57,58].

### 2.3. Methods of Data Collection

#### 2.3.1. QST Assessments

Thermal stimuli were delivered by using a computerized thermal stimulator TSA II (Medoc, Ramat Yishay, Israel), with a standard 30 × 30 mm TSA II thermode, onto the ventral surface of the nondominant forearm of the volunteer (Figure 1). Care was taken to ensure the best contact between the probe and the forearm surface. Sessions were conducted by a previously trained researcher and held in a closed-door room, with a controlled temperature, to avoid interruptions. All subjects declared themselves thermally comfortable at the beginning of the experiment. Before initiation of the testing, participants received a detailed explanation of the process and signed an informed consent. The stimulus intensity rating was obtained through the 11-point numeric rating scale (NRS) [20,59,60].

#### 2.3.2. QST Protocol

Based on the literature [20,45,48,51,53,61,62,63,64,65,66,67] and group meetings, including research physicians and anesthesiologists, an initial protocol was defined. This protocol was piloted in the members of the team and in volunteers with chronic pain, other than those included in the present study. Minor adjustments were performed, namely to move the test site from the hand to the ventral surface of the forearm, not using cold thresholds, and decrease temperature time from 2 °C/s to 8 °C/s. The protocol comprised 3 data collection:Heat Pain Threshold: Application of a baseline temperature (32 °C), which gradually increased at 1 °C/s to a maximum of 50 °C and decreased at 8 °C/s. Volunteers were asked, 6 times, to indicate the transition point at which the nonpainful warm sensation changed into a painful heat sensation, by pressing a button that stopped the temperature-increasing process and electronically recording it. The mean of the responses to the last 3 stimuli was taken as the “heat pain threshold” [45,51,53,62,64].Suprathreshold Pain Magnitude: Subjects were asked to report the level of pain intensity 9 times during three different random 3-s contact heat stimuli. The 3 different temperatures were the mean pain of the first test, a neutral temperature of 36 °C and a suprathreshold temperature of 49 °C. Volunteers were asked to rate discomfort and/or pain in NRS (0 meaning no pain to 10 meaning the worst pain they can imagine) [12,59,60,63,67].Tonic Pain Magnitude: A constant temperature of 47 °C was applied for 1 min. Volunteers were asked to report pain in NRS, if any, at 10, 20, 40, and 55 s [48,65,66].

The subject was not able to look at the computer screen during the tests. The three tests were performed in the sequence mentioned above, with an interval of at least 1 min [62]. All patients referred that they were not sensitive from the previous test before moving to the next one.

### 2.4. Statistical Analysis

The sampling process was based on previous studies with QST that included a range from 14 to 51 subjects [19,21,68,69]. Despite the difficulties in recruiting volunteers, we exceed our expectations of 30 participants per group. A total of 90 participants were enrolled in this study (40 healthy volunteers and 50 patients with chronic pain).

Descriptive statistics were used to summarize the demographic and clinical characteristics of the sample. First, data were tested for normal distribution, applying Kolmogorov–Smirnov test or Shapiro–Wilk. SPSS 27 software and a 95% confidence level were used to perform an independent sample T test (or Mann–Whitney U) to compare both groups for each QST quantitative variable. The Chi-square test was used to make comparisons among categorical variables. The proportion estimates and the respective confidence intervals were calculated.

MesH terms were used to divide the sample age into 3 groups: young (<18 years), adults (18–65 years), and elderly (>65 years) subjects [70]. Based on the literature, reported pain was analyzed through Numeric Rating Scale (NRS) classification and also grouped, to facilitate the presentation of the results, as follows: 0 in NRS as no pain, 1–3 in NRS as mild pain, 4–6 in NRS as moderate pain, and 7–10 in NRS as severe pain [71,72]. In order to simplify the interpretation of the results regarding the location of the pain, this variable was split into two categories: musculoskeletal pain and head, throat, and other (such as perianal) pain.

Taking into account the classification found in the literature regarding the application of the BPI questionnaire, 7 was established as a “cutoff” between the low and high impact of pain on activity, sleep, and humor of the subjects [73,74]. The variable “treatment of pain” was interpreted according to the WHO analgesic ladder [75]: “no analgesic treatment”, “step 1—nonopioid”, “step 2—weak opioid”, and “step 3—strong opioid”.

## 3. Results

The sample size comprised 90 volunteers with a mean age of 38.7 ± 24.9 years, of which 69.2% were adults, 26.2% elderlies, and 4.6% young (considering missing age data for 25 subjects of the chronic pain group). The control group, healthy young students with a mean age of 19.9 ± 1.1 years, included mainly adults (92.5%) and a few young (7.5%) volunteers. The chronic pain group (CPG), with a mean age of 68.7 ± 10.7 years (*p* < 0.001), had adults (32.0%) and mainly elderly (68.0%) volunteers. Due to legal aspects related to access to the data collection, half of the chronic group age information was missing. The control group had initially 50 subjects; however, 10 volunteers, who suffered from chronic pain, were excluded (Table 1).

Related to gender, there was no significant difference between the groups (*p* = 0.984). Nonetheless, the female gender represents 65.6% of the total sample and is also predominant in both groups (60% in the control group and 70% in the chronic pain group) (Table 1).

The chronic pain group reported significantly more depression than the control group (*p* = 0.001). Seven patients in the chronic pain group reported considering themselves depressed, with four diagnosed with depression. Furthermore, four chronic pain patients were taking medication for depression. However, due to the high number of missing values in the chronic pain group, no further conclusions can be achieved.

Only eight subjects in the control group had felt occasionally acute mild pain (50% musculoskeletal pain and 50% head, throat, or perianal pain), with a duration of less than a month. In the control group, no subjects reported pain at the moment of the tests. Nobody in the control group reported pain at its minimum (Table 1).

The pain was categorized as musculoskeletal in all chronic pain patients. According to the inclusion criteria, chronic pain was only evaluated for CPG: knee pain (44.0%), hip pain (24.0%), and nonspecific joint pain (22.0%). Additionally, 66.0% of CPG subjects reported pain during the interview, being classified with moderate intensity for 32.0%, the mean pain intensity was 3.58 ± 3.15, and 18.0% reported severe pain. On average, the CPG subjects reported pain intensity of 5.14 ± 2.12, and 48.0% classified it as “moderate pain”. Chronic pain was “severe” at its maximum in 58.0% of CPG subjects, the mean maximum pain intensity being 6.76 ± 2.31. Overall, 8% of the CPG classified the pain as “severe” even at its minimum, and the mean minimum pain was 2.58 ± 2.65 (Table 1).

The Shapiro–Wilk test (<0.05) and visual inspection of histograms, Q-Q plots, and interquartile diagrams showed the absence of a normal distribution for these variables in both groups. Although there are some Z score values between −1.96 and +1.96, the Shapiro–Wilk test was considered more accurate [76], and, therefore, nonparametric tests were performed considering nonnormal distributions for both groups. Thus, both groups were compared using the Mann–Whitney U statistical test. Statistically significant differences (*p* < 0.01) were demonstrated with respect to the history of pain (pain at the time of the interview, average pain, and maximum pain).

No significant differences were found between groups for pain interference categories concerning general activities, sleep, or humor. However, the percentages and visual inspection of the bar charts indicate a higher interference in the CPG (Figure 2). For example, 91.7% of subjects with a high impact of pain on their quality of sleep suffered from chronic pain. Furthermore, comparing the means of pain interference, the differences were statistically significant concerning humor (0.043) but not with respect to general activity and sleep (*p* = 0.644 and *p* = 0.873, respectively). Nonetheless, the CPG group assigned higher values of interference for the three variables than the control group (Table 1).

In general, there were no subjects taking strong opioids and only six subjects of the CPG reported the use of weak opioids. There were significant differences in the analgesic medication used by the individuals with chronic pain, of which 68.9% resorted to nonopioid analgesics.

For the QST analysis, since the variables “mean pretest threshold” (1st–3rd stimuli) and “mean threshold” (4th–6th stimuli) did not follow a normal distribution (*p* < 0.001, Kolmogorov–Smirnov test), nonparametric tests were used. The Wilcoxon test demonstrated significant differences (*p* < 0.001) between the “mean pretest threshold” (43.83 ± 3.54 °C in the control group and 47.76 ± 2.44 °C in the CPG) and the “mean threshold” among all the subjects, showing higher trigger temperatures in the 4th–6th stimuli. The comparison between the study groups revealed that the “mean pain threshold” was significantly higher in the CPG (*p* < 0.01, Mann–Whitney U test, Table 2, Figure 3). There were no differences between males (46.79 ± 3.14 °C) and females (46.99 ± 3.31 °C) in this variable (Mann–Whitney U, *p* = 0.6192).

There was an increase in the pain intensity for higher temperatures in the suprathreshold test (Table 2). For all three variables (“suprathreshold mean temperature”, “sham suprathreshold at 36 °C”, and “suprathreshold at 49 °C”), the Kolmogorov–Smirnov and Shapiro–Wilk tests rejected the null hypothesis, thus demonstrating that they did not present a normal distribution. Friedman test including pairwise analysis demonstrated significant differences (*p* < 0.01). When using Mann–Whitney U tests to compare these variables between study groups, it was verified that there was only a significant difference in “threshold mean temperature”, reaching higher values in CPG (4.97 ± 2.98) than in the control group (3.04 ± 2.76). The same tendency occurred for the “sham suprathreshold at 36 °C” without significant differences. Nonetheless, the NRS values for “suprathreshold at 49 °C” were closer (Table 2, Figure 4). There were no differences between males (4.26 ± 2.68 °C) and females (4.07 ± 3.25 °C) in suprathreshold mean temperature (Mann–Whitney U, *p* = 0.6897), between males (0.32 ± 0.59 °C) and females (0.56 ± 1.19 °C) in sham suprathreshold at 36 °C (Mann–Whitney U, *p* = 0.9414), and between males (6.52 ± 2.03 °C) and females (6.47 ± 2.82 °C) in suprathreshold at 49 °C (Mann–Whitney U, *p* = 0.6739).

The four moment values of the tonic pain test were not normally distributed (*p* < 0.01 in Kolmogorov–Smirnov and Shapiro–Wilk tests). The results showed differences in the perception of pain through the continuous stimulus, with the values of NRS increasing from 10 to 55 s during tonic stimulus (Friedman test). However, this was only statistically significant between the NRS at 20 and 55 s (*p* = 0.011). When studying each group separately, this pattern is maintained in the control group, whereas in the CPG there were only statistically significant differences between the beginning and end of the stimulus (10 and 55 s, *p* = 0.045). Through visual inspection of the bar charts for each group (Figure 5), it is possible to notice a change between groups after the initial period. In CPG, it is more noticeable tendency for higher NRS pain values after 20 s (Table 2). Nevertheless, there were no statistically significant differences in tonic pain perception between the two groups. There were no differences between males (4.07 ± 2.22 °C) and females (4.14 ± 3.14 °C) at 10 s of tonic stimulation (Mann–Whitney U, *p* = 0.9627), between males (4.07 ± 2.37 °C) and females (3.97 ± 2.98 °C) at 20 s of tonic stimulation (Mann–Whitney U, *p* = 0.8316), between males (4.32 ± 2.64 °C) and females (4.27 ± 3.05 °C) at 40 s of tonic stimulation (Mann–Whitney U, *p* = 0.9289), and between males (4.61 ± 2.45 °C) and females (4.49 ± 3.10 °C) at 55 s of tonic stimulation (Mann–Whitney U, *p* = 0.8350).

## 4. Discussion

The main results of this controlled clinical trial demonstrated that the proposed QST protocol is suitable for different groups of people, with or without chronic pain. This study protocol was based on several studies for the thermal pain threshold [45,51,53,62,64], suprathreshold pain magnitude [12,59,60,63,67], and tonic pain [48,65,66]. These three different tests were used to include different points in pain experience and have different values in predicting the subsequent clinical pain, with the most consistent results regarding the heat pain threshold [52,77].

Previous studies found lower thresholds for pain in chronic pain patients, including other QST modalities such as pressure pain thresholds [30,78] that can be related to the different protocols used. Conversely, other studies reported higher thresholds in patients with chronic pain, other than OA [77,79,80,81]. A previous study, using QST, identified higher pain susceptibility phenotypes in knee OA that were more likely to develop persistent knee pain [44]. In most studies, participants with OA appear to be more sensitive to pain both at the affected joint and at unaffected sites compared to nonpain controls. Interestingly, when OA was unilateral, both extremities showed hyperalgesia [40,42,82]. It has been proposed that this may result from peripheral nerve sensitization caused by inflammatory mediators [12]. Additionally, it was found that athletes have the highest tolerance to thermal pain when compared to patients suffering from chronic musculoskeletal pain [41]. These results suggest that several resources should be used to evaluate and better characterize pain, improving treatment with a more comprehensive and individualized approach.

In our study, the CPG presented significantly higher heat pain thresholds, demonstrating hypoesthesia rather than hyperalgesia which was initially expected. This fact can be related to factors such as emotion and cognition, which were not evaluated in detail in the present study. However, there was a significantly higher pain perception for the threshold temperature in the CPG. This suggests that, although these patients presented a higher tolerance to heat, after reaching the pain threshold, they demonstrated hyperalgesia, which may be explained by the central sensitization phenomena that occur in the chronic pain [83,84]. The results for suprathreshold pain perception were only significant for threshold temperature, indicating that this might be the level of heat stimulus most useful to chronic pain patients, avoiding excessive overstimulation of higher temperatures. On the other hand, other studies found a correlation between postoperative pain and preoperative sensibility to suprathreshold rather than threshold temperatures [12,53]. Differences between populations and timings (preoperative and postoperative) may justify these findings. It should also be considered that more than 50% of CPG patients reported moderate to severe pain and were subjected to higher threshold temperature values than control subjects.

Many studies using QST were based in the German Research Network on Neuropathic Pain (DFNS), which has taken efforts to set up a protocol with a good comparability of data across different laboratories [25,45]. However, the DFNS protocol was unable to identify heat hypoalgesia and used a small group of healthy participants with a larger age range (17 to 75 years old). It also differs from the present protocol on the locations assessed (face, hand, and foot), which is relevant since QST reference values are region-dependent. Furthermore, in more exposed regions, such as hands and face, there are probably high variations among people, whereas more protected areas, such as the forearm, may be the ideal place for QST tests. In fact, we notice in our pilot study (QST in nondominant hand) a huge difference between those who worked as cooks, for example, who did not feel heat pain and others whose hands are not exposed to heat daily, which prompted us to move from hand to forearm to perform the tests in the present study.

By evoking a physiologically representative array of stimuli, QST is capable of detecting both negative and positive phenomena (ranging deficits from hypoesthesia to allodynia and hyperalgesia), becoming critical for the understanding of pain physiology, mainly when compared to scales that assess self-reported symptoms [20,39]. The protocol proposed in our study can be effective in testing hypoesthesia to allodynia by using the limits test and hyperalgesia through the suprathreshold test in patients with chronic pain, while tonic pain, despite being a useful test [48,65,66], presented very similar results in both groups evaluated in the present study.

In the current literature, studies that evaluated the response to tonic pain are less common, but chronic pain patients are reported to have earlier sensitization [65]. In our sample, the CPG showed a similar pain perception during the first 10 s of stimulus, demonstrating a tendency to hyperalgesia after 20 s, although these were residual differences without statistically significant differences. The fact that the only significant differences in NRS were detected between the beginning of the stimulus and the end is consistent with the expected increase in pain during a longer stimulation and can also be influenced by other variables, including the emotional component of the pain. It is also interesting that, unlike CPG, the control group, after a period of steady stimulation (20 and 40 s), felt a decrease in pain perception. This is probably due to a refractory period of thermal receptors after a prolonged stimulation leading to a transient reduction in depolarization [62,66]. Temporal summation occurs when multiple stimuli of the same intensity culminate in pain due to neuroplasticity mechanisms [85] which may occur in our study, consistent with the subject’s feedback that repetition of the same stimulus in suprathreshold tests leads to an increase in sensitivity towards the end of the test [13,86]. Although the “pretest threshold” is not a reliable measure, it was significantly lower in comparison to the three last stimuli, also highlighting this effect.

The development of this protocol aimed to provide a shorter and more clinically applicable tool to characterize how patients experience pain, overcoming the complexity and time consumption of previous protocols. However, some obstacles remain to be addressed, such as training health care providers and access to equipment (ideally bedside availability) [22]. It should also be noted that all the tests in this study were performed in identical conditions, which is seldom possible in clinical practice, affecting reproducibility.

This study has limitations. The fact that all the measurements (questionnaires and QST tests) were performed in the same session, with no follow-up period between the intervention and the outcomes, characterizing the cross-sectional nature of the study design, does not allow for causal inferences. Furthermore, there is the possibility of depression being induced by chronic pain, introducing a confounding factor in this causality relation, that is, in fact, bidirectional [87,88]. Other limitations of the present study included a nonprobabilistic convenience sampling due to logistical accessibility. Therefore, no sample size calculation methods were used due to the difficulty in defining the effect size that should be labeled clinically relevant in external validity. However, since our population was reasonably homogeneous due to strict inclusion criteria, a smaller sample is required for a small variance [89,90]. Moreover, we obtained a sample size larger than several similar studies conducted previously, enhancing the statistical power of our results, and internal validity [19,21,68,69]. Besides, the CPG, although homogeneous, was made up only of patients from one of the largest hospitals in Portugal. This might introduce an external validity bias since we probably selected patients in more severe conditions than the ones receiving care in smaller hospitals or primary health care facilities.

Heterogeneity and lack of reproducibility are problems inherent in QST studies, as the results may not be consistent for the same individual over a period of time [28]. In this study, the same modality and location of testing were used in all the subjects, although the timing of testing was an internal validity limitation. Previous studies repeated the tests with a gap of 1 to 4 weeks; hence, in our study, it could have been beneficial to perform more than one measurement per subject to increase reliability and detect changes in pain perception over time [16,41]. We tried to test all CPG patients after surgery, but only a very few were able to be contacted due to data protection, and none attended to repeat the QST tests due to lack of motivation. Nevertheless, QST measurements, and particularly the method of limits, are time-dependent and may improve with practice. Taking one test at one specific moment in time may have reduced this time bias. Regarding the heterogeneity of age between groups, it was recently proposed that age differences might not play a significant role in some pain regulatory processes [41]. Difficulties in defining a truly “healthy” test group for pain studies have been discussed, and a thorough screening of this particular group is necessary to avoid a possible bias [91]. Nevertheless, we believe that QST will be very useful to guide patient care in the future, extending and improving standard neurological examination at bedside.

## 5. Conclusions

Together with other tools, QST can help to characterize the complexity of the biopsychosocial model of pain. Despite its great potential, this equipment is still costly and currently unavailable to most primary clinicians. In our study, we found that chronic OA patients, in the preoperative period, were more tolerant to heat stimulus than healthy young volunteers to pain thresholds (hypoesthesia). However, once the pain was evoked, chronic pain patients demonstrated higher sensibility (hyperalgesia). Suprathreshold and tonic stimulation did not prove to be very helpful tests to differentiate chronic pain patients from healthy subjects.

## Figures and Tables

**Figure 1 biomedicines-11-01023-f001:**
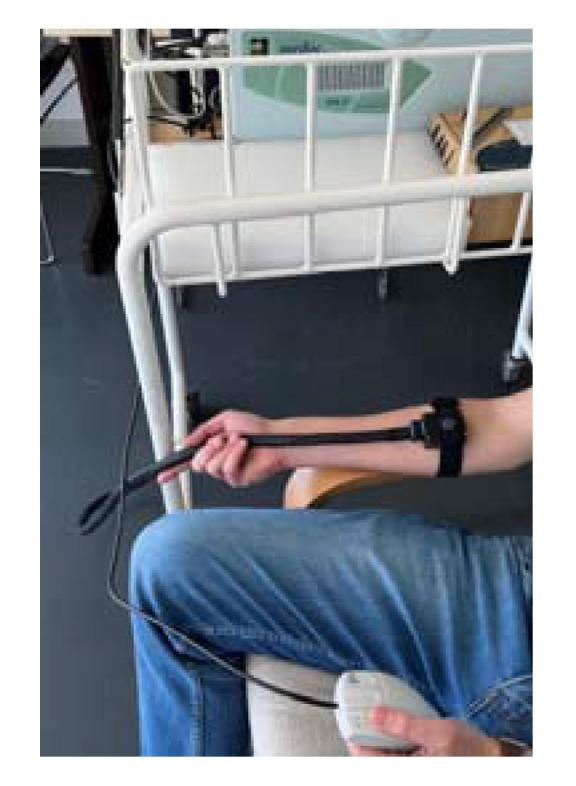
Picture of the TSA II thermode in position for the QST assessments.

**Figure 2 biomedicines-11-01023-f002:**
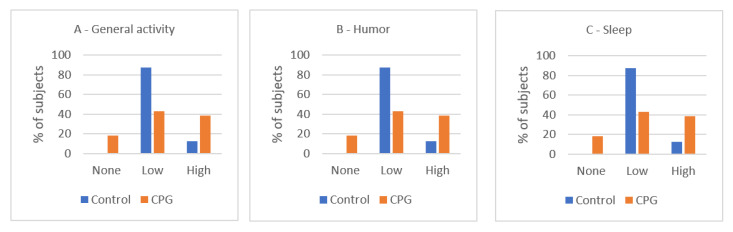
Impact of pain in the control group and CPG (% of subjects per level of interference). (**A**) Level of interference with general activity, (**B**) level of interference with humor, and (**C**) level of interference with sleep. CPG, chronic pain group. Visual inspection of the bar charts indicates a higher interference in the CPG (no statistical differences in Chi-square test).

**Figure 3 biomedicines-11-01023-f003:**
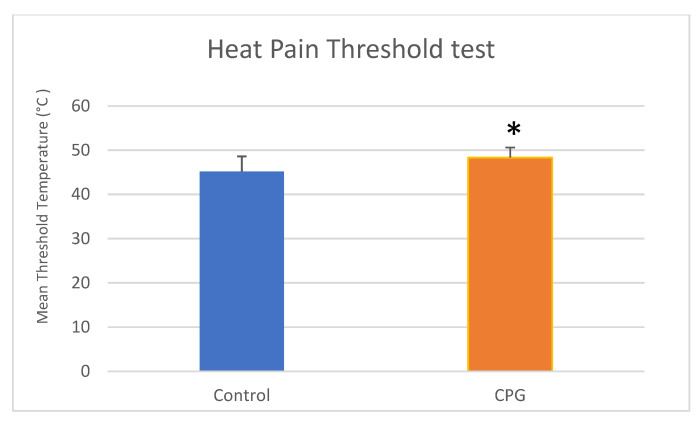
Comparison of mean ± SD heat pain threshold temperatures (°C) between the control group and the CPG using Mann–Whitney U test (* *p* < 0.05). SD, standard deviation; CPG, chronic pain group. There was significant difference between groups (Mann–Whitney U test).

**Figure 4 biomedicines-11-01023-f004:**
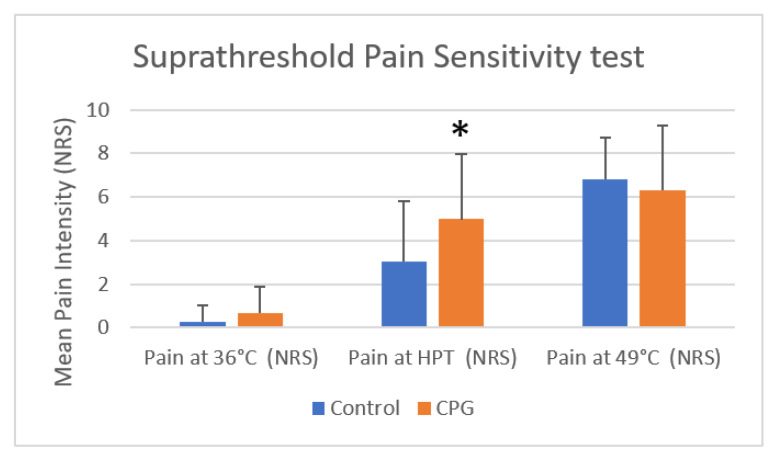
Comparison of mean ± SD pain sensitivity in suprathreshold tests in the control group and the CPG (NRS at 36 °C, mean heat pain threshold, and 49 °C) using Mann–Whitney U test (* *p* < 0.05). HPT, heat pain threshold; NRS, numeric rating scale; SD, standard deviation. The results for the suprathreshold tests demonstrated significant differences between the control group and the CPG at HPT but not at 36 °C or 49 °C (Mann–Whitney U test).

**Figure 5 biomedicines-11-01023-f005:**
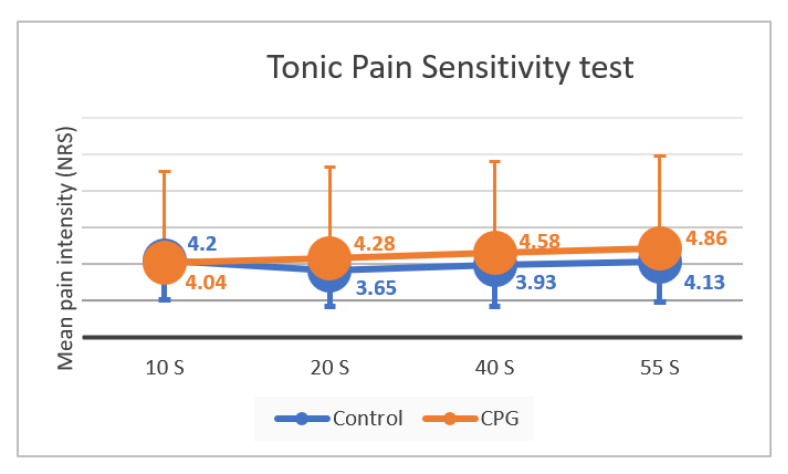
Intensity of pain perception throughout the tonic stimulus (10, 20, 40, and 55 s) in the control group and in the CPG, with standard deviation bars (mean ± SD). NRS, numeric rating scale; SD, standard deviation; CPG, chronic pain group. There were no significant differences in NRS during the tonic stimulus between the control group and the CPG (Friedman test).

**Table 1 biomedicines-11-01023-t001:** Comparative analysis between the control group and the chronic pain group (CPG) (*n* = 90).

Variable	Control (*n* = 40)	CPG (*n* = 50)	*p* * Value
1—Females *n* (%)	(24) 60.0	(35) 70.0	0.984
2—Age distribution *n* (%)			
Young (<18 years)	3 (7.5)	0 (0.0)	0.001
Adults (18–65 years)	37 (92.5)	8 (32.0)	
Elderly (>65 years)	0 (0.0)	17(68.0)	
3—Predominant pain location *n* (%)			
Musculoskeletal	4 (50.0)	50 (100.0)	0.001
Others	4 (50.0)	0 (0.0)	
4—Pain intensity in the interview *n* (%)			
No pain	40 (100.0)	17 (34.0)	0.001
Mild pain	0 (0.0)	5 (10.0)	
Moderate pain	0 (0.0)	19 (38.0)	
Severe pain	0 (0.0)	9 (18.0)	
Mean (NRS ± SD)	0.00 ± 0.00	3.58 ± 3.15	0.001
5—Average of past pain intensity *n* (%)			
No pain	32 (80.0)	0 (0.0)	0.001
Mild pain	1 (2.5)	12 (24.0)	
Moderate pain	6 (15.0)	24 (48.0)	
Severe pain	1 (2.5)	14 (28.0)	
Mean (NRS ± SD)	0.98 ± 2.06	5.14 ± 2.12	
6—Pain at its maximum *n* (%)			
No pain	32 (80.0)	0 (0.0)	0.001
Mild pain	0 (0.0)	5 (10.0)	
Moderate pain	5 (12.5)	16 (32.0)	
Severe pain	3 (7.5)	29 (58.0)	
Mean (NRS ± SD)	1.23 ± 2.56	6.76 ± 2.32	0.001
7—Pain interference on the subjects’ activity *n* (%)			
None	1 (12.5)	5 (10.2)	0.254
High	1 (12.5)	21 (42.9)	
Low	6 (75.0)	23 (46.9)	
Mean (NRS ± SD)	3.75 ± 2.32	5.80 ± 2.79	0.644
8—Pain interference on the subjects’ humor *n* (%)			
None	0 (0.0)	9 (18.4)	0.060
High	1 (12.5)	19 (38.8)	
Low	7 (87.5)	21 (42.9)	
Mean (NRS ± SD)	2.50 ± 2.14	5.35 ± 3.40	0.043
9—Pain interference on the subjects’ ability to sleep *n* (%)			
None	5 (62.5)	13 (26.5)	
High	1 (12.5)	11 (22.4)	0.127
Low	2 (25.0)	25 (51.0)	
Mean (NRS ± SD)	2.25 ± 3.28	3.88 ± 3.31	0.873

*n*, number; CPG, chronic pain group; NRS, numeric rating scale; SD, standard deviation of the mean. * Chi-square, Mann–Whitney U, and t tests were used in these comparisons, as appropriate.

**Table 2 biomedicines-11-01023-t002:** Comparative analysis of the QST variables between the control group and the chronic pain group (CPG) (*n* = 90).

Variable	Control (*n* = 40)	CPG (*n* = 50)	*p* * Value
1—Threshold mean temperature, T ± SD (°C)	45.19 ± 3.43	48.32 ± 2.29	0.001
2—Perception of pain intensity at 36 °C, NRS ± SD	0.28 ± 0.73	0.67 ± 1.19	0.162
3—Perception of pain at threshold mean temperature, NRS ± SD	3.04 ± 2.76	4.97 ± 2.98	0.003
4—Perception of pain intensity at 49 °C, NRS ± SD	6.80 ± 1.92	6.28 ± 2.99	0.792
5—Perception of pain at 10 s of tonic stimulation, NRS ± SD	4.20 ± 2.13	4.04 ± 3.33	0.694
6—Perception of pain at 20 s of tonic stimulation, NRS ± SD	3.65 ± 1.98	4.28 ± 3.26	0.323
7—Perception of pain at 40 s of tonic stimulation, NRS ± SD	3.93 ± 2.26	4.58 ± 3.31	0.362
8—Perception of pain at 55 s of tonic stimulation, NRS ± SD	4.13 ± 2.19	4.86 ± 3.31	0.203

T, temperature; NRS, numeric rating scale for pain intensity; SD, standard deviation. * Mann–Whitney U and Kruskal–Wallis tests were used in these comparisons as appropriate.

## Data Availability

The data that support the findings of this study are available from the corresponding author, P.D., upon reasonable request.

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
