# Peer review of "Outcomes of a QST Protocol in Healthy Subjects and Chronic Pain Patients: A Controlled Clinical Trial"

_biomedicines, 2023, doi:10.3390/biomedicines11041023_

Round 1

Reviewer 1 Report

The authors prospectively evaluated the thermal sensation in 40 healthy subjects and 50 patients with chronic pain to develop a new, reproducible and less time consuming thermal quantitative sensory testing (QST). The results suggested that the new heat threshold QST was helpful in evaluating hypoesthesia and that the sensitivity threshold temperature test demonstrated hyperalgesia in individuals with chronic pain. 

This is an important study suggesting the usefulness of a new thermal QST method. As many clinical trials of therapeutic agents for pain will be conducted in the future, it may also be available as a surrogate endpoint in these trials. Therefore, this manuscript will attract a broad range of readers from physicians to researchers. 

Although I do not have any critical comments, minor issues to strengthen this manuscript are raised as follows: 

1. The background of patients with pain, such as sex and etiology, should be detailed. 

2. Please reconfirm the use of abbreviations. For example, “NRS” in Table 1 should be explained in the footnote of this table. 

3. Findings obtained from each figure should be briefly described in the legend of each figure. 

Author Response

Porto, 21 March 2023

Dear Editor-in-Chief of Biomedicines,

Re: Ref. biomedicines-2272084

Thank you for the comments regarding our paper “Outcomes of a QST protocol in healthy subjects and in chronic pain patients: a controlled clinical trial”. We have reviewed the manuscript according to the critiques raised by the reviewers and thank them for their comments.

We hope you will consider the revised version of the manuscript suitable for publication in Biomedicines. All alterations in the manuscript are easily recognizable, since they are highlighted in yellow. A point-by-point answer to the issues raised by reviewers can be found below.

Reviewer 1

The authors prospectively evaluated the thermal sensation in 40 healthy subjects and 50 patients with chronic pain to develop a new, reproducible and less time consuming thermal quantitative sensory testing (QST). The results suggested that the new heat threshold QST was helpful in evaluating hypoesthesia and that the sensitivity threshold temperature test demonstrated hyperalgesia in individuals with chronic pain.

This is an important study suggesting the usefulness of a new thermal QST method. As many clinical trials of therapeutic agents for pain will be conducted in the future, it may also be available as a surrogate endpoint in these trials. Therefore, this manuscript will attract a broad range of readers from physicians to researchers.

Although I do not have any critical comments, minor issues to strengthen this manuscript are raised as follows:

  1. The background of patients with pain, such as sex and etiology, should be detailed.

Answer: Thank you very much for your time reviewing this manuscript, your kind comments, and your valuable suggestions. Related to your request, we clarified it in lines 108-110: “All patients presented chronic pain from knee or hip due to osteoarthritis and were approached in the pre-operative period for knee or hip replacement.” Related to gender it is already described in the text lines 193-195 and we also included detailed information in the first line of table 1.

  1. Please reconfirm the use of abbreviations. For example, “NRS” in Table 1 should be explained in the footnote of this table.

Answer: We agree and proceeded to these corrections in the manuscript.

  1. Findings obtained from each figure should be briefly described in the legend of each figure.

Answer: We agree and proceeded to these descriptions in the manuscript figures 2,3,4 and 5.

Reviewer 2 Report

Thank you for permitting me to review this manuscript 

Material and methods: please add a photo or picture of the thermal stimulator II

Statistical analysis 

the authors describe previuos study explaining the number of participants , but they should elaborate d which were the expectations .

results , table 1 need some realignement  n°6,7

table 1 : only 2 out of 8 values were significantly different between groups ,  authors should check litterature if these two values can be detected otherwise , if yes , therefore this particular QST may have less value , this should be discussed 

Line 338 please provide some reference or additional fact for this affirmation 

Line362 this sentence id difficult to understand please rephrase it 

I suggest using a special abbreviation for thz QST used in this study as  there is other QST described by authors used in other types of pain , therefore this might be subject to confusion 

conclusion 

the authors should be cautious with their finding and  suggestions should not be included in conclusion such as line 400 401 , which might be included in the discussion only not in conclusion 

Author Response

Porto, 21 March 2023

Dear Editor-in-Chief of Biomedicines,

Re: Ref. biomedicines-2272084

Thank you for the comments regarding our paper “Outcomes of a QST protocol in healthy subjects and in chronic pain patients: a controlled clinical trial”. We have reviewed the manuscript according to the critiques raised by the reviewers and thank them for their comments.

We hope you will consider the revised version of the manuscript suitable for publication in Biomedicines. All alterations in the manuscript are easily recognizable, since they are highlighted in yellow. A point-by-point answer to the issues raised by reviewers can be found below.

Reviewer 2

Thank you for permitting me to review this manuscript

Material and methods: please add a photo or picture of the thermal stimulator II

Answer: Thank you very much for your time reviewing this manuscript, and your valuable suggestions. We add a photo (figure 1) in the manuscript of the TSA II demonstrating how the test was performed.

Statistical analysis

the authors describe previous study explaining the number of participants, but they should elaborate d which were the expectations.

Answer: The sampling process was done by convenience, since the recruitment of volunteers is a difficult process, our expectation was to include at least 30 patients per group to examine the feasibility of the QST tests and exceed the sample size of previous studies. Nevertheless, we reached 90 volunteers. We add this information in lines 161-162: “Despite the difficulties in recruiting volunteers, we exceed our expectations of 30 participants per group”

table 1 : only 2 out of 8 values were significantly different between groups ,  authors should check litterature if these two values can be detected otherwise , if yes , therefore this particular QST may have less value , this should be discussed

Answer: Table 1 summarizes the analysis that was performed in accordance with the literature and previous studies. In the first manuscript draft we had 120 references (and we read a lot more than that for the protocol development and groups definition) that we shortened to make it easier to read and understand. None of the previous studies could direct us in other way than that described. The limitations of the study are discussed in detail including your pertinent concern in lines 393-425. Finally, there are 9 significantly differences between groups in table 1 and not only 2, and the objective was not to detect differences, but demonstrate that the groups were different by nature.

Line 338 please provide some reference or additional fact for this affirmation

Answer: We provided some references and improved the statement “while tonic pain, despite being a useful test [48, 65, 66], presented very similar results in both groups evaluated in the present study.”

Line362 this sentence id difficult to understand please rephrase it

Answer: The sentence was rephrased.

I suggest using a special abbreviation for thz QST used in this study as there is other QST described by authors used in other types of pain, therefore this might be subject to confusion

Answer: We understand this request, it will be useful for a less attentive reader. However, this will look like we are inventing a new modality of QST, and this is not true. Thus, we prefer to keep in the way it is to follow the previous literature in the area avoiding misunderstandings.

conclusion

the authors should be cautious with their finding and suggestions should not be included in conclusion such as line 400 401 , which might be included in the discussion only not in conclusion

Answer: We agree with you, the sentence in question was moved to the end of the discussion in the manuscript, as recommended.

Reviewer 3 Report

This study has proposed a new, reproducible, and less time-consuming thermal QST protocol to help characterize and monitor pain. Also, it compared QST outcomes between healthy and chronic pain subjects. 40 healthy young healthy medical students and 50 chronic pain patients were evaluated in individual sessions comprising pain history, followed by QST assessments on pain threshold, suprathreshold, and tonic pain. The findings showed that in the chronic pain group, the pain threshold was higher (hypoesthesia) and pain sensibility was also higher (hyperalgesia) compared to healthy participants. However, sensitivity to the suprathreshold and tonic stimulus did not differ between the groups. The heat threshold in the QST tests showed to help evaluate hypoesthesia and the sensitivity threshold temperature test can demonstrate hyperalgesia in individuals with chronic pain.

The authors are encouraged to consider the following points for the revisions:

·       In Fig 1 are there any statistical tests applied to find a statistically significant difference or not? Please elaborate.

·       Please clarify the heat pain threshold and thermal pain threshold.

·       Data are presented as Mean and SD. Does this mean that the data distribution was normal? Which test was used to check the normal data distribution? Please add. Otherwise use non-parametric.

·       It is difficult to understand what comparison was exactly done in relation to age. university students age and chronic pain patients age. This point is also remarkable to be clarified in the abstract. what about gender related differences? Did the authors find any dominant response in either gender or age X gender interaction?

·       Please add the limitations of this study and points related to internal and external validity.

Author Response

Porto, 21 March 2023

Dear Editor-in-Chief of Biomedicines,

Re: Ref. biomedicines-2272084

Thank you for the comments regarding our paper “Outcomes of a QST protocol in healthy subjects and in chronic pain patients: a controlled clinical trial”. We have reviewed the manuscript according to the critiques raised by the reviewers and thank them for their comments.

We hope you will consider the revised version of the manuscript suitable for publication in Biomedicines. All alterations in the manuscript are easily recognizable, since they are highlighted in yellow. A point-by-point answer to the issues raised by reviewers can be found below.

Reviewer 3

This study has proposed a new, reproducible, and less time-consuming thermal QST protocol to help characterize and monitor pain. Also, it compared QST outcomes between healthy and chronic pain subjects. 40 healthy young healthy medical students and 50 chronic pain patients were evaluated in individual sessions comprising pain history, followed by QST assessments on pain threshold, suprathreshold, and tonic pain. The findings showed that in the chronic pain group, the pain threshold was higher (hypoesthesia) and pain sensibility was also higher (hyperalgesia) compared to healthy participants. However, sensitivity to the suprathreshold and tonic stimulus did not differ between the groups. The heat threshold in the QST tests showed to help evaluate hypoesthesia and the sensitivity threshold temperature test can demonstrate hyperalgesia in individuals with chronic pain.

The authors are encouraged to consider the following points for the revisions:

  • In Fig 1 are there any statistical tests applied to find a statistically significant difference or not? Please elaborate.

Answer: Thank you very much for your time reviewing this manuscript, your kind comments, and your valuable suggestions. Related to your request, the interference of pain with general activity, humor and sleep was analyzed comparing means between the two groups (results in Table 1), using a Mann- Whitney U test, demonstrating significant differences in pain interference with the humor. In order to simplify the interpretation of results, these variables were subcategorized in “None”, “Low” and “High” interference. Using the Chi Square test, significant differences were found between the two groups regarding the categories for these 3 variables of interference. To improve the manuscript, we follow your concern, and we added the used tests in the legend of all figures with graphics.

  • Please clarify the heat pain threshold and thermal pain threshold.

Answer:  We changed thermal pain threshold to heat pain threshold in 2.3.2. QST protocol. Now it is standard in the entire manuscript.

  • Data are presented as Mean and SD. Does this mean that the data distribution was normal? Which test was used to check the normal data distribution? Please add. Otherwise use non-parametric.

Answer:  As described in the methods section (lines 165-166): “First, data were tested for normal distribution applying Kolmogorov-Smirnov test.”. Additionally, Shapiro-Wilk test was analyzed and we added in the methods. When we found a normal distribution, we performed a t test to compare the variables (which was the case only for the level of pain interference in general activity and sleep). In the other variables, we did not obtain a normal distribution and, therefore, applied a Mann-Whitney U, Chi square and Kruskal Wallis and Friedman test, as adequate and described in the manuscript.

  • It is difficult to understand what comparison was exactly done in relation to age. university students age and chronic pain patients age. This point is also remarkable to be clarified in the abstract. what about gender related differences? Did the authors find any dominant response in either gender or age X gender interaction?

Answer: This study aimed to test the feasibility of the proposed protocol in 2 different populations: 1 – healthy young/adult and 2 -  adult/elderly chronic pain patients. This was related to the inclusion criteria allowing to test our protocol in two different populations. The age comparisons were made only to certify that the populations were different by nature. We clarify It better in the abstract.

There were no differences between genders. Nevertheless, we added the gender comparisons for QST variables in the results on pages 7 and 8 (lines: 250-252, 263-267, 279-284). Additionally, as mentioned in the result section “the female gender represents 65.6% of the total sample and is also predominant in both groups (60% in control group and 70% in chronic pain group).” Therefore, we can assume that the differences found in the QST tests were not remarkably influenced by gender, since homogeneity regarding this demographic variable was demonstrated in our sample.

  • Please add the limitations of this study and points related to internal and external validity.

Answer: The limitations of the study are explored in the discussion section, particularly starting in lines 393-425. Despite not explicit the text, the information is there. To clarify it better we highlighted the points related to internal and external validity, as requested (lines: 401, 405, 407, 412).

Reviewer 4 Report

Thank you for the opportunity to review this manuscript. The rationale is sound as well as the methodological framework.

I have just a couple of minor concerns:

- why did the Authors select only an upper threshold  temperature (i.e. high temperature) and not a lower one (i.e. low temperature), at the same time.

- how were ensured the security standards regarding the protocol (burns, etc).

-unit on y axis in Fig. 2 is missing (°C ?)

Author Response

Porto, 21 March 2023

Dear Editor-in-Chief of Biomedicines,

Re: Ref. biomedicines-2272084

Thank you for the comments regarding our paper “Outcomes of a QST protocol in healthy subjects and in chronic pain patients: a controlled clinical trial”. We have reviewed the manuscript according to the critiques raised by the reviewers and thank them for their comments.

We hope you will consider the revised version of the manuscript suitable for publication in Biomedicines. All alterations in the manuscript are easily recognizable, since they are highlighted in yellow. A point-by-point answer to the issues raised by reviewers can be found below.

Reviewer 4

Thank you for the opportunity to review this manuscript. The rationale is sound as well as the methodological framework.

I have just a couple of minor concerns:

- why did the Authors select only an upper threshold temperature (i.e. high temperature) and not a lower one (i.e. low temperature), at the same time.

Answer: Thank you very much for your time reviewing this manuscript, your kind comments, and your valuable suggestions. Related to your request, we select only an upper threshold temperature because it was proved to be the best thermal QST option in the several previous studies  (Verdugo 1992- PMID: 1628207 , Hilz 1999- PMID: 10576229, Agostinho 2009- PMID: 19019713, Sand 2010- PMID: 20656701, Heldestad 2010- PMID: 20478739, Yarnitsky 2012- PMID: 22480803, Granot 2003: PMID: 12766652, Werner 2003- PMID: 14576553, Werner 2004- PMID: 14695732, Lautenbacher 2005- PMID: 15876494, Pan 2006- PMID: 16508387, Strulov 2007- PMID: 17113350, Martinez 2007- PMID: 17717244, Rudin 2008-PMID: 18477083, Yarnitsky 2008- PMID: 18079062, Schestatsky 2011- PMID: 22297885, Eisenberg 2010- PMID: 20621419, Lautenbacher 2010- PMID: 20850220). Cold pain threshold has a large variance in the data limits individual analyses (Moloney 2011- PMID: 21826684) and didn’t demonstrate to be beneficial for our protocol. During the pilot study some volunteers reported relief or fresh sensation with cold thresholds. Finally, we aimed to propose a “reproducible and less time consuming thermal QST protocol”. We add this information in Line 138.

- how were ensured the security standards regarding the protocol (burns, etc).

Answer: The safety of the subjects was assured to them when they signed the informed consent that was written accordingly to the equipment manual and previous QST protocols using computerized thermal stimulator TSA II. The patients were carefully instructed to press the button to stop the stimulus (now depicted in the new figure 1) whenever heat sensation became a pain sensation or pain-like discomfort. They were also informed that they can interrupt the tests in any time and withdrawn if they want. Furthermore, even at maximum temperature (50 Celsius degrees) the few seconds in higher temperatures were not enough to cause any kind of burns or skim damage. Detailed information can be found in lines 135-153.

-unit on y axis in Fig. 2 is missing (°C ?)

Answer: We proceeded to this correction in the manuscript.

Round 2

Reviewer 2 Report

the authors have responded to my queries